

# Inconsistent year-to-year fluctuations limit the conclusiveness of global higher education rankings for university management

Johannes Sorz[1], Bernard Wallner[2,3,4], Horst Seidler[2] and Martin Fieder[2]

[1] Office of the Rectorate, University of Vienna, Vienna, Austria
[2] Department for Anthropology, University of Vienna, Vienna, Austria
[3] Unit for Quality Assurance, University of Vienna, Vienna, Austria
[4] Department for Behavioral Biology, University of Vienna, Vienna, Austria

## ABSTRACT

**Backround.** University rankings are getting very high international media attention, this holds particularly true for the Times Higher Education Ranking (THE) and the Shanghai Jiao Tong University's Academic Ranking of World Universities Ranking (ARWU). We therefore aimed to investigate how reliable the rankings are, especially for universities with lower ranking positions, that often show inconclusive year-to-year fluctuations in their rank, and if these rankings are thus a suitable basis for management purposes.

**Methods.** We used the public available data from the web pages of the THE and the ARWU ranking to analyze the dynamics of change in score and ranking position from year to year, and we investigated possible causes for inconsistent fluctuations in the rankings by the means of regression analyses.

**Results.** Regression analyses of results from the THE and ARWU from 2010–2014 show inconsistent fluctuations in the rank and score for universities with lower rank positions (below position 50) which lead to inconsistent "up and downs" in the total results, especially in the THE and to a lesser extent also in the ARWU. In both rankings, the mean year-to-year fluctuation of universities in groups of 50 universities aggregated by descending rank increases from less than 10% in the group of the 50 highest ranked universities to up to 60% in the group of the lowest ranked universities. Furthermore, year-to-year results do not correspond in THES- and ARWU-Rankings for universities below rank 50.

**Discussion.** We conclude that the observed fluctuations in the THE do not correspond to actual university performance and ranking results are thus of limited conclusiveness for the university management of universities below a rank of 50. While the ARWU rankings seems more robust against inconsistent fluctuations, its year to year changes in the scores are very small, so essential changes from year to year could not be expected. Furthermore, year-to-year results do not correspond in THES- and ARWU-Rankings for universities below rank 50. Neither the THES nor the ARWU offer great value as a tool for university management in their current forms for universities ranked below 50, thus we suggest that both rankings alter their ranking procedure insofar as universities below position 50 should be ranked summarized only in groups of 25 or 50. Additionally, the THE should omit the

Corresponding author
Martin Fieder,
martin.fieder@univie.ac.at

peer reputation survey, which most likely contributes heavily to the inconsistent year-to-year fluctuations in ranks, and ARWU should be published less often to increase its validity.

## INTRODUCTION

Global higher education rankings have received much attention recently and, as can be witnessed by the growing number of rankings being published every year, this attention is not likely to subside. Besides the arguable use of ranking results as an instrument for university management, it is still a common practice in many universities to use rankings as an indicator for academic performance. Rankings became a big business and as of today a plethora of regional and national rankings exist, advocated by their publishers as potentially efficient and effective means of providing needed information to universities on areas needing improvement (*Dill & Soo, 2005*). Numerous studies have analyzed and criticized higher education rankings and their methodologies (*Van Raan, 2005*; *Buela-Casal et al., 2007*; *Ioannidis et al., 2007*; *Hazelkorn, 2007*; *Aguillo et al., 2010*; *Benito & Romera, 2011*; *Hazelkorn, 2011*; *Rauhvargers, 2011*; *Tofallis, 2012*; *Saisana, d'Hombres & Saltelli, 2011*; *Safon, 2013*; *Rauhvargers, 2013*; *Bougnol & Dula, 2014*). This casts justified doubt on a sensible comparison of universities hailing from different higher education systems and varying in size, mission and endowment based on mono-dimensional rankings and league tables and hence on the usability of such rankings for university management and policy making (*O'Connell, 2013*; *Hazelkorn, 2014*). Several studies have demonstrated that data used to calculate ranking scores can be inconsistent. Bibliometric data from international databases (Web of Science, Scopus), used in most global rankings to calculate research output indicators, favor universities from English-speaking countries and institutions with a narrow focus on highly-cited fields, which are well covered in THE databases. This puts universities from non-English-speaking countries, with a focus on the arts, humanities and social sciences, at a disadvantage when being compared in global rankings (*Calero-Medina et al., 2008*; *Van Raan, Leeuwen & Visser, 2011*; *Waltman et al., 2012*). Data submitted by universities to ranking agencies (e.g., personnel data, student numbers) can be problematic to compare due to different standards. These incompatibilities are being amplified because university managers have become increasingly aware of global rankings and try to boost their performance by "tweaking" the data they submit to the ranking agencies (*Spiegel Online, 2014*). Beyond all the data issues, there is the effect that universities with lower ranking positions often encounter volatile ups and downs in their consecutive year-to-year ranks. These effects make university rankings an inconclusive tool for university managers: the ranking results simply do not reflect the universities' actual performance or their management strategies. Ranking results need to be consistent to be

of use, so that long-term strategies (e.g., the hiring of high-calibre researchers from abroad or investments in doctoral education) are reflected in year-to-year scores and ranks and in perennial trends. Furthermore, results from various rankings should be concordant to allow a sort of meta-analysis of rankings.

*Bookstein et al. (2010)* found unacceptably high year-to-year variances in the score of lower ranked universities in the THE, *Jovanovic et al. (2012)* and *Docampo (2013)* found a large number of fluctuations and inconsistencies in the ranks of the ARWU. As we again observed puzzling results in the THE 2014–15 and the ARWU 2014 that were both published recently, we accordingly analyzed the fluctuations in score and rank of the THE and the ARWU. By calculating regression analyses for both rankings for consecutive years for 2010–2014, we tried to determine the amount of inconsistent fluctuations that can most likely not be explained by changes in university performance (e.g., by increase of publications/citations, change in student/faculty numbers) and listed the universities with the most extreme changes in ranking position for the THE and ARWU. Furthermore, the mean percentages of universities that changed their rank in their groups were calculated for groups of 50 universities aggregated by descending rank in both THE and ARWU, and we calculated a regression of the ranking positions of the first 100 universities in the THE 2014 on the first 100 universities in the ARWU 2014.

## THE

The methodology of the THE was revised several times in varying scale, before and after the split with Quacquarelli Symonds (QS) in 2010 and the new partnership with Thompson Reuters. THE calculates 13 performance indicators, grouped into the five areas Teaching (30%), Research (30%), Citations (30%), Industry income (2.5%) and International outlook (7.5%). However, THE does not publish the scores of individual indicators, only those of all five areas combined. Since 2010, the research output indicators are calculated based on Web of Science data. Most of the weight in the overall score is made up by the normalized average citations per published paper (30%), and by the results of an academic reputation survey (33%) assessing teaching and research reputation and influencing the scores of both areas (*Rauhvargers, 2013*; *Times Higher Education, 2014*). In the past, criticism has been levied against this survey. Academic peers can choose universities in their field from a preselected list of institutions and, although universities can be added to the list, those present on the original list are more likely to be nominated. This leads to a distribution skewed in favor of the institutions at the top of the rankings (*Rauhvargers, 2011*; *Rauhvargers, 2013*). THE allegedly addressed this issue by adding an exponential component to increase differentiation between institutions, yet no information is available on its mode of calculation (*Baty, 2011*; *Baty, 2012*).

## ARWU

ARWU ranks more than 1,000 (of ca. 17,000 universities in the world) and publishes the best 500 on the web. In addition, ARWU offers to field rankings that cover several subjects and subject rankings for Mathematics, Physics, Chemistry, Computer Science and Economics & Business. Universities are ranked according to their research performance,

including alumni (10%) and staff (20%) winning Nobel Prizes and Fields Medals, highly cited researchers in 21 broad subject categories in the Web of Science (20%), papers published in Nature and Science (20%), papers indexed in major citation indices (20%), and the per capita academic performance of an institution (10%). Calculation of indicators remained relatively constant since 2004. ARWU ranks universities individually or into bands by sorting on the total score, which is the linearly weighted sum of the six research output indicator scores derived from the corresponding raw data by transformations. Institutional data (number of academic staff) is not provided by universities but obtained from national agencies such as ministries, national bureaus and university associations (*ARWU, 2013*). In contrast to the THE, there are no teaching/student related indicators or any peer survey component in the ARWU. Due to reliance on ISI subject fields, the areas of natural sciences, medicine and engineering dominate the citation indicator, putting universities with a focus on the arts, humanities and social sciences. The per capita performance is the only ARWU indicator that takes into account the size of the institution, thus small but excellent institutions have less of a chance to perform well in the ARWU-Ranking (*Rauhvargers, 2011*). Already several studies, i.e., *Docampo (2011)* and *Docampo (2013)* analyzed the ARWU and its indicators and found inconsistencies and unwanted dynamical effects. We have no further information beside the public available information on the indicators and their weights on how the scores are calculated for THE and ARWU.

## METHODS

We used the publicly available data on scores and ranks from the THE and ARWU for the years 2010, 2011, 2012, 2013 and 2014, including in the THE all universities ranked between 1 and 200 and in the ARWU the universities ranked between 1 and 100, as ARWU starts aggregating the ranking from rank 101 on. We performed the following analysis for both rankings: (i) we plotted and regressed the scores of the rankings of the year $t-1$ on the scores of the year $t$; (ii) we plotted and regressed the ranks of the rankings of the year $t-1$ on the ranks of the year $t$; (iii) we plotted the associations of scores and ranks and approximated the function of the association between scores and ranks; (iv) we investigated the concordance (ranking position of the first 100 universities) of the THE ranking with the ARWU ranking. For this purpose, we regressed the position of the first 100 universities in the THE-Ranking (2014–15) on the ranking position of the first 100 university in the ARWU-Ranking (2014) and finally (vii) to include also universities ranked below 200 we aggregated the THE from "position 1" on in steps of 50 universities, i.e., we defined 8 aggregated ranking groups (1–50, 51–100, 101–150, 151–200, 201–250, 251–300, 301–350, 351–400) for the THE. As the ARWU starts aggregating in steps of 100 universities from rank 201 on, we refrained from including universities ranked lower than 200 in the ARWU in our analysis, due to comparability; therefore, for the ARWU we defined 4 aggregated ranking groups of also 50 universities (1–50, 51–100, 101–150, 151–200).

On basis of this rearrangement, we made the following calculations: we calculated the year-to-year mean fluctuations (%), thus the percentage of universities that changed their

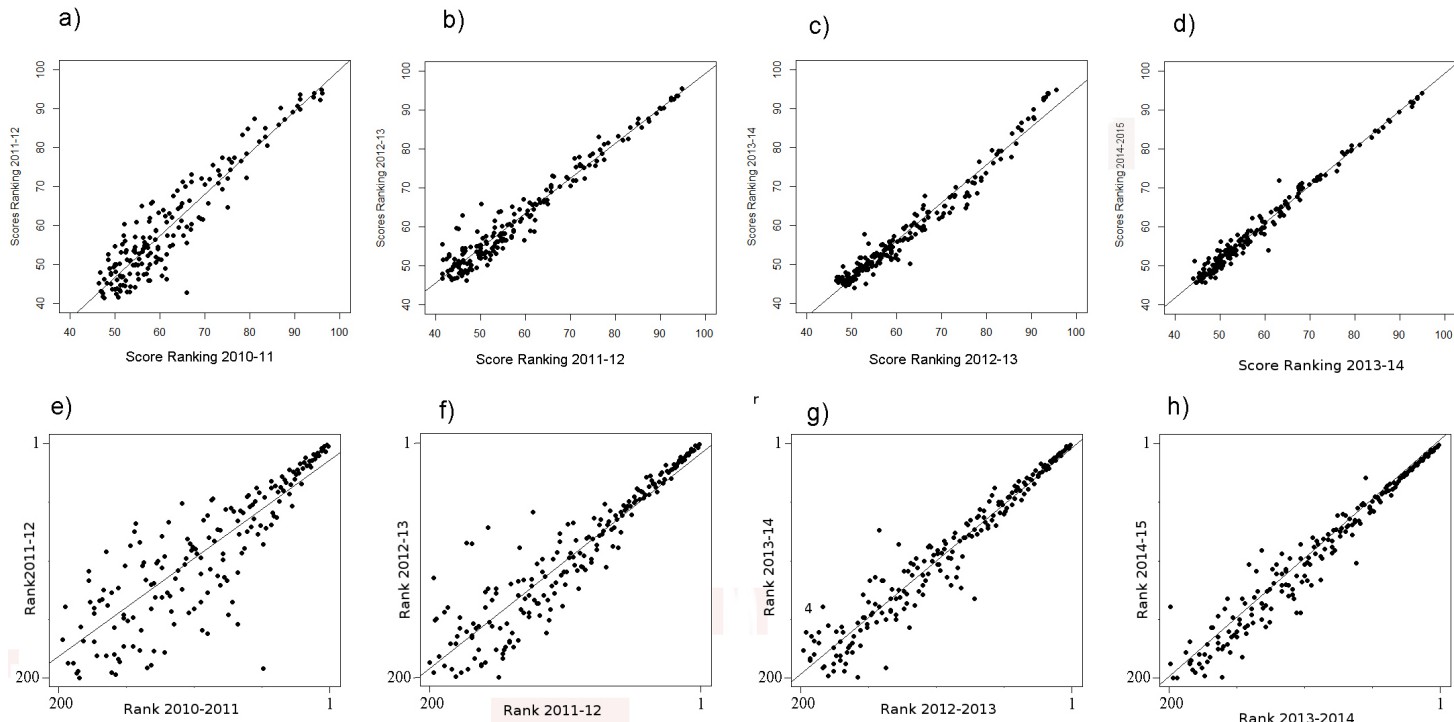

**Figure 1** (A–H) THE: scores of the year $t$ regressing on the score of the year $t-1$ from the ranking 2010–11 on. (A–D) Ranks of the year $t$ regressing on the ranks of the year $t-1$ from the ranking 2010–11 on (E–H) . Linear regression line indicates perfect association, e.g., no changes in ranks and scores between two consecutive rankings. Remark: THE denotes the rankings in academic years (Figs. 1 and 2), whereas ARWU denotes the rankings in calendar years (Figs. 4 and 5).

rank beyond their aggregated ranking groups (moving upwards, moving downwards, being newly in the rank, respectively dropping out of the ranking) to get an estimate of yearly fluctuation according to the ranking groups. We calculated the mean change of years 2012, 2013 and 2014. Direction and amount of change were not considered; only the fact that a university did change ranking group, i.e., moving within the ranking, dropping out of the ranking, respectively coming newly into the ranking, was counted.

## RESULTS

### THE regression of the scores and ranks of two consecutive years

The regression of the scores—particularly of the ranking 2010–2011 regressing on the scores of the ranking of 2011–2012—shows very high fluctuations (Fig. 1A), especially for the lower ranked universities. Moreover, the fluctuations among the lower ranked universities seem to be higher compared to the THE performed by QS before 2010 (*Bookstein et al., 2010*, Fig. 1). Note that in the rankings in the years following 2010–2011, the fluctuations in the THE-Ranking did decrease (Figs. 1B–1D). Tables S1A–S1H show the regression models including regression coefficients, confidence intervals, $p$-values, degrees of freedom and $R^2$ values for each model. $R^2$ ranges from 0.71 to 0.98, i.e., the models with the small $R^2$ do explain less variance. This confirms the visual impression of particularly high year-to-year fluctuations as depicted in Figs. 1A and 1E. Also when

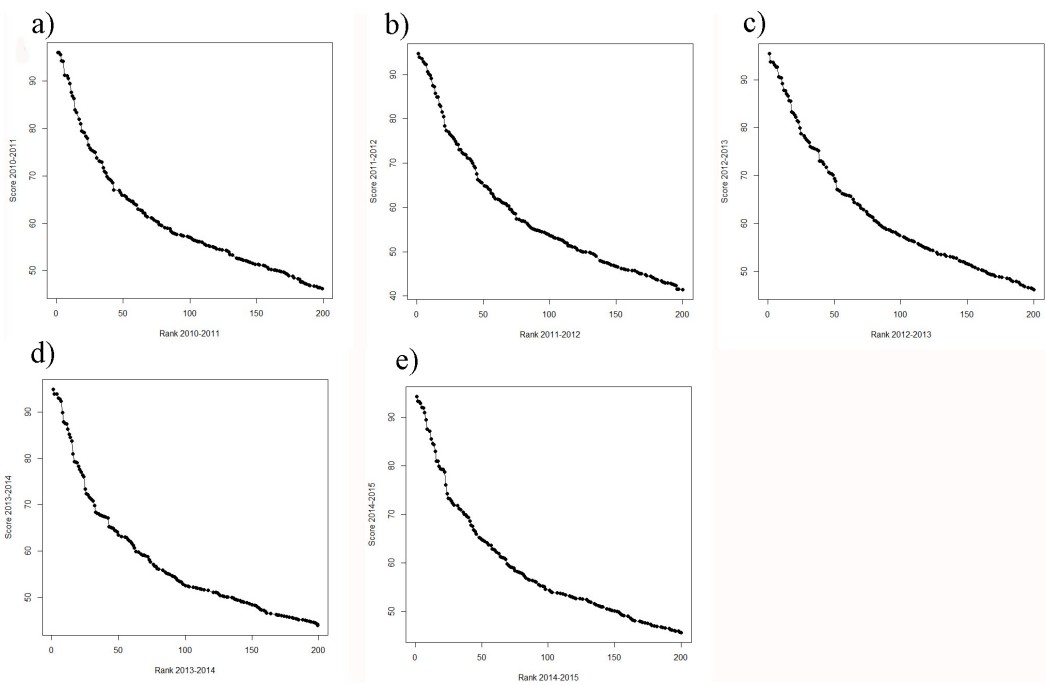

**Figure 2** (A–E) Ranks plotted against scores for the THE (A) score 2010–2011 plotted against rank 2010–2011; (B) score 2011–2012 plotted against rank 2011–2012; (C) score 2012–2013 plotted against rank 2012–2013; (D) score 2013–2014 plotted against rank 2013–2014 and (E) score 2014–2015 plotted against rank 2014–2015.

universities are aggregated in groups of 50 universities by decreasing rank, the observed fluctuations increase with increasing rank from year to year and the mean year-to-year fluctuation (mean percentages of universities that changed their rank in their group) increases in each subsequent lower ranked group (Fig. 3). While the mean year-to-year fluctuation is less than 10% in the first group (1–50), it is over 60% in the lowest ranked universities group (351–400). The most extreme cases of "fluctuating university ranks" from 2010 to 2014 in the THE rankings are displayed in Table S3.

### Association between scores and ranks

A general problem of the THE remains: the difference in the scores among the 50 highest scoring universities is considerably higher compared to the difference among the lower scoring universities. This clearly suggests a non-linear relationship between scores and ranks (Figs. 2A–2E). The consequence is that the ranks of the high scoring universities are much more robust to deviations in the scores from year to year. In the lower ranking universities, however, even very small, more or less random deviations (around 0.5%) lead to unexpected "high jumps" in the ranks from year to year (Figs. 1E–1H). We assume, that these fluctuations are to a large extent caused by the results of the peer reputation survey, which can be skewed by low response rates (*Rauhvargers, 2011*) and the "Matthew effect" (*Merton, 1968*). Interestingly, the association between scores and ranks approximates pretty well a power function of the form rank = score + score$^b$, there b ranges between −4.106 and −4.88 for the THE ranking. In Fig. S1, the fit on basis of a power function for

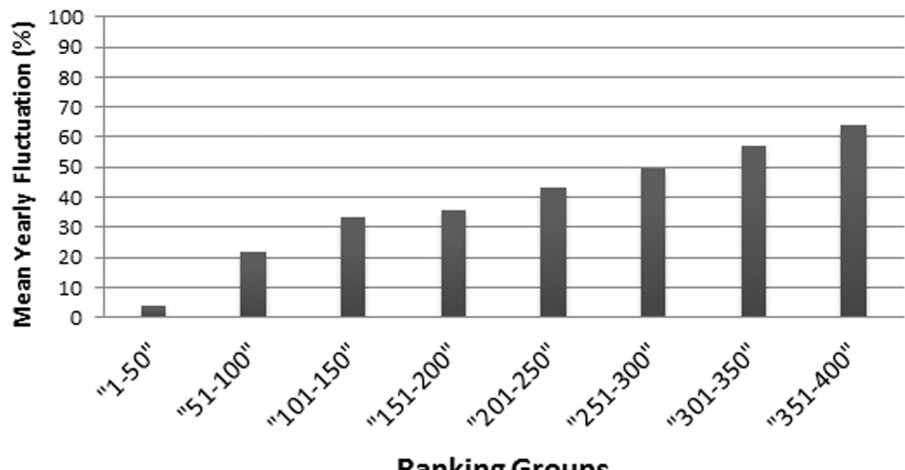

**Figure 3** **Mean year-to-year fluctuation (mean of the years 2012, 2013 and 2014) in the THE (%), according to ranking groups.** For example the percentage of Universities within a ranking group of 50 universities, that changed position beyond the ranking group (moved up, or moved down, respectively dropped out of the ranking).

the ranking 2014 is shown exemplarily (as the figures are quite similar we did refrain from plotting all the power fits).

## ARWU

While still on a high level, the regression of ranks and scores of the ARWU, show much less fluctuations compared with the THE. This indicates a more robust set of indicators. Furthermore the ARWU shows a similar, but even a more extreme pattern of non-linearity between ranks and scores, compared with the THE. Particularly the first ranked university, Harvard University, scores far ahead of all the other universities in the ARWU at each year. As in the THE the association between ranks and scores flattens from rank of 50 on (Figs. 4A–4E). As in the THE the non-linear relationship between ranks and scores increases the fluctuations in the ranking positions of the universities ranked approximately below 50 from year to year (Figs. 3A–3D). Tables S2A–S2H show the regression models including regression coefficients, confidence intervals, $p$-values, degrees of freedom and $R^2$ values for each model. $R^2$ ranges from 0.71 to 0.98, i.e., the models with the small $R^2$ do explain less variance, the amount of inconsistent fluctuations seem to be higher. The higher $R^2$ values indicate, that the ARWU shows less inconsistent fluctuations compared to the THE, which confirms the visual impression of the Figs. 3A–3H and confirms our notion that the inconsistent fluctuations in the THE are largely caused by the results of the peer reputations survey, which is not included in the ARWU. As in the THE the associations scores and ranks approximates pretty well a power function of the form rank = score + score$^b$, there b ranges between −2.96 and −3.00 for the ARWU (compared with −4.106 and −4.88 for the THE).

As in the THE, if universities are aggregated in steps of 50 universities by decreasing rank, the fluctuations increase with increasing rank from year to year and the mean

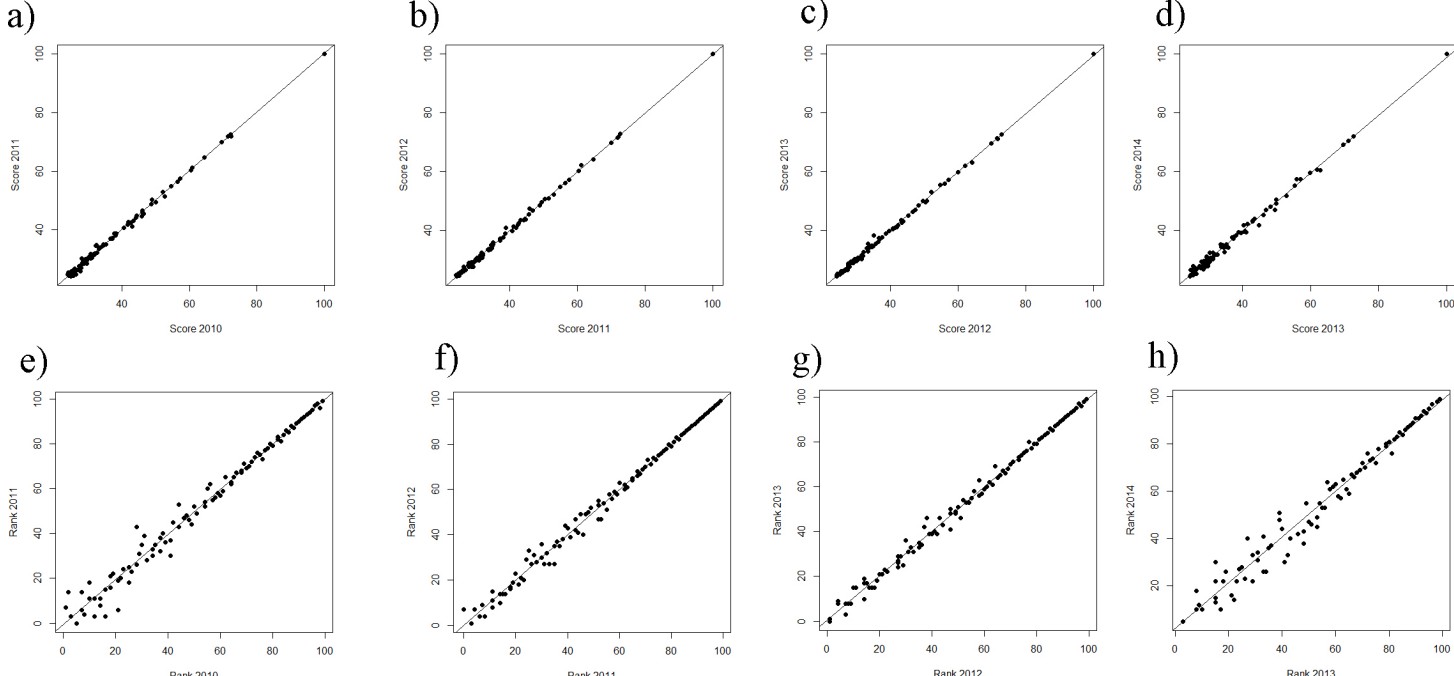

**Figure 4 (A–D) ARWU: scores of the year $t$ regressing on the score of the year $t − 1$ from the ranking 2010 on.** (A–D) Ranks of the year $t$ regressing on the ranks of the year $t − 1$ from the ranking 2010 on (E–H). Linear regression line indicates perfect association, e.g., no changes in ranks and scores between two consecutive rankings.

year-to-year fluctuation (%) increases in lower ranking groups (Fig. 6). While the mean year-to-year fluctuation is less than 10% in the first group (1–50) it is over 40% in the lowest ranked universities group (151–200). The most extreme cases of "fluctuating university ranks" from 2010 to 2014 in the ARWU rankings, are displayed in Table S4.

### Correlation between THE and ARWU

A really dramatic amount of inconsistent fluctuations reveal the regression of the ranks in the THE on the ranks in the ARWU: for the universities ranked approximately lower than the 50th rank, there is virtually no correlation between the THE and the ARWU (Fig. 7). Regression could only be plotted for universities ranked in both rankings among the first 100. The $R^2$ of 0.52 indicates, as seen in the plot (Fig. 7), that only a relatively small amount of the variance can be explained by the association between the THE and the ARWU (Table S3). i.e., the inconsistent fluctuations are quite high. This implies that the THE and ARWU make substantial different statements on the performance of universities ranked below 50, thus making both rankings hard if not impossible to compare. This effect is to some degree understandable, due to the different setup of indicators in both rankings; however, one could assume that the reflection of academic performance of universities is reflected more homogenously in both rankings. More homogenous results would also allow a "meta-ranking" which could be of more value for university management than singular contradicting ranking results.

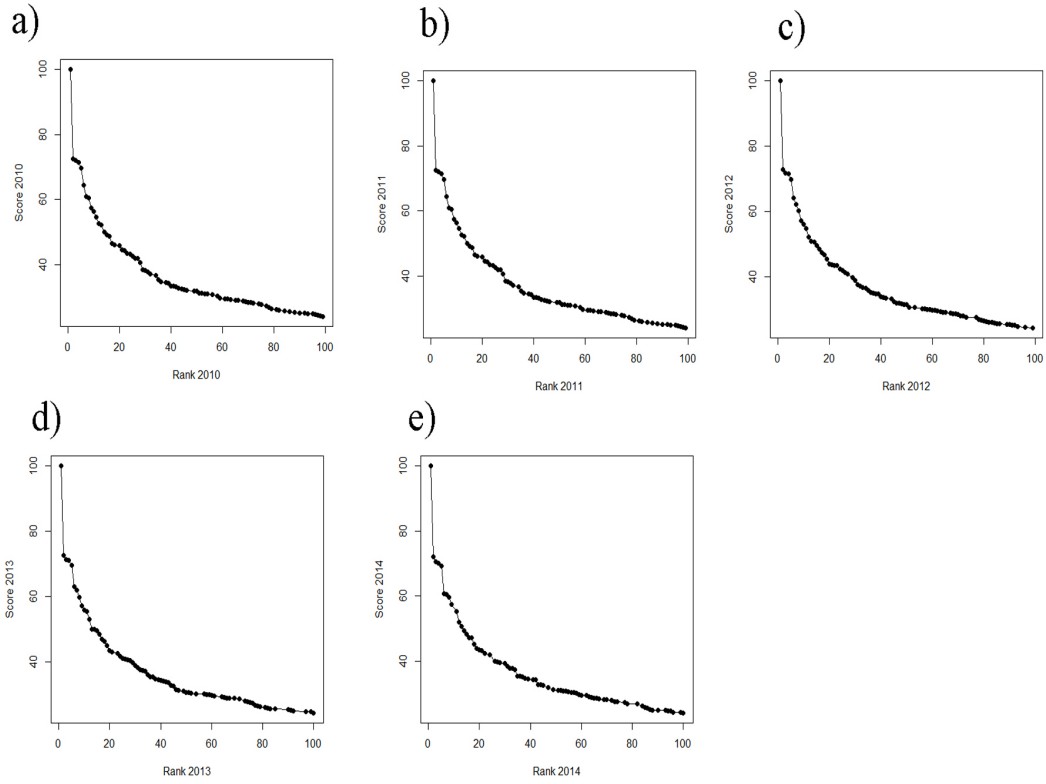

**Figure 5** **(A–E) Scores vs. ranks for the ARWU from 2010 to 2014.** (A) score 2010 plotted against rank 2010; (B) score 2011 plotted against rank 2011; (C) score 2012 plotted against rank 2012; (D) score 2013 plotted against rank 2013 and (E) score 2014 plotted against rank 2014.

## DISCUSSION

High ranking positions achieved by a small group of universities are often self-perpetuating, especially due to the intensive use of peer review indicators, which improve chances of maintaining a high position for universities already near the top (*Bowman & Bastedo, 2011*; *Rauhvargers, 2011*). This phenomenon also corresponds to the "Matthew effect," which was coined by *Merton (1968)* to describe how eminent scientists will often get more credit than a comparatively unknown researcher, even if their work is similar: credit will usually be given to researchers who are already famous. The intensive and exaggerated discussion in the media of the "up and downs" of universities in the THE is particularly misleading for universities with lower ranking positions (below approximately a score of 65% and a rank of 50; above scores of 65%, the relationship between ranks and scores is steeper, and it flattens for scores below 65%). This is because the ranking positions suggest substantial shifts in university performance despite only very subtle changes in score. In fact, merely random deviations must be assumed. One reason lies in the weighing of indicators by THE, with the emphasis on citations (30% of the total score) and the peer reputation survey (33% of the total score). For lower ranked universities, a few highly cited publications, or the lack thereof, or few points asserted by peers in the reputation survey, probably make a significant difference in total score and position.

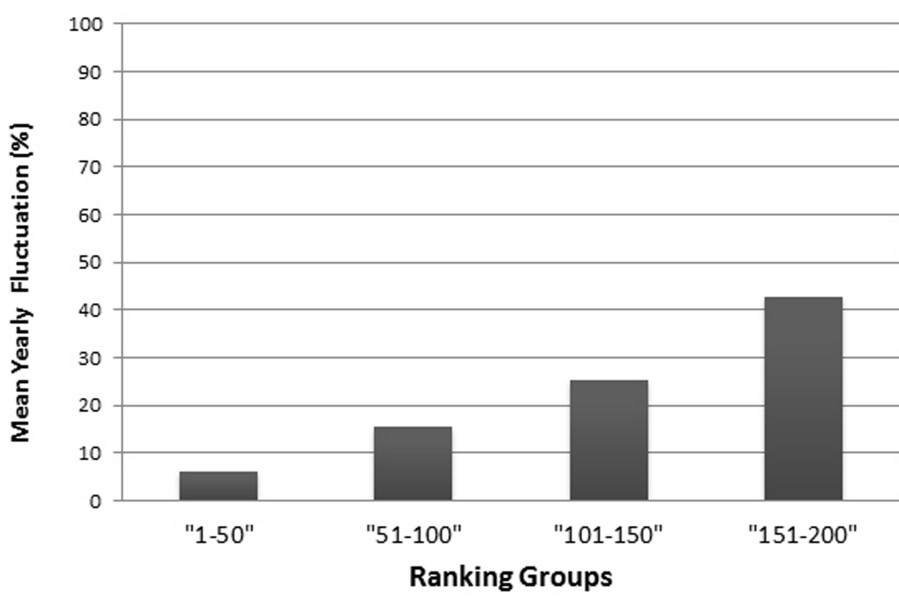

**Figure 6 Mean year-to-year fluctuation (mean of the years 2012, 2013 and 2014) in the ARWU (%), according to ranking groups.** E.g., the percentage of Universities within a ranking group of 50 universities, that changed position beyond the ranking group (moved up, or moved down, respectively dropped out of the ranking).

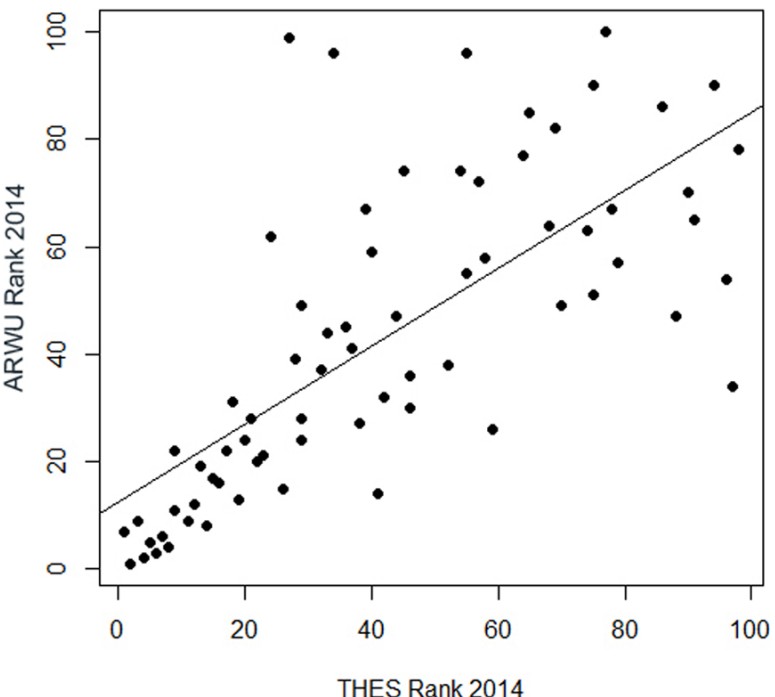

**Figure 7 Regression of the ranking positions of the first 100 universities in the THE 2014 on the first 100 universities in the ARWU 2014 ($n = 71$ universities).**

Ranking results have a major influence on the public image of universities and can even impact their claim to resources (*Espeland & Sauder, 2007*; *Hazelkorn, 2011*). Accordingly, inconsistent fluctuations in ranking positions can have serious implications for universities, especially when the media or stakeholders interpret them as direct results of more or less successful university management. The use of monodimensional rankings for university management is generally doubtful. Our results show that THE, especially in its current form, has very limited value for the management of universities ranked below 50. This is because the described fluctuations in rank and score probably do not reflect actual performance, whereby the results cannot be used to assess the impact of long-term strategies. "Rankings are here to stay, and it is therefore worth the time and effort to get them right" warns *Gilbert (2007)*. What could be done to address the fluctuations in the THE for universities below rank 50 to make it a more usable tool to assess actual performance for university management? THE has already addressed fluctuations to some extent by ranking universities only down to position 200, followed by groups of 25 from 201–300 and groups of 50 from 300 to 400. Nonetheless, based on our data we believe that this is not going far enough and suggest that universities should be summarized in groups of 25 or 50 below the position of 50. Furthermore, we believe that these inconsistent fluctuations are caused to a large extent by the results of the peer reputation survey, which can be skewed by low response rates (*Rauhvargers, 2011*) and biased by the "Matthew effect" that favours already well-renowned institutions (*Merton, 1968*). The latter could possibly also help to explain the consistency in the scores and ranks in the group of the top-50 universities. Thus, in order to increase the validity of the THE, the peer reputation survey should be omitted or given less weight in future rankings.

The analysed curves of scores vs. ranking positions in Fig. 2 do have analogous characteristics for example to semi-logarithmic curves produced in analytic biochemistry. The accuracy of such curves is limited to the steepest slope of the curve, whereas asymptote areas deliver higher fuzziness (*Chan, 1992*). Thus, a further suggestion to avoid the blurring dilemma is the methodological approach of introducing a standardization process for THE data. This would involve using common suitable reference data to create calibration curves represented by non-linearity or linearity. Simple calibration would be as suggested to categorize universities in ranking groups or to apply a transformation to the scores, such as a logarithmical transformation. Comparing the year to year fluctuation in the ARWU with the THE reveals, that fluctuation in the ARWU ranking is overall lower as in the THE ranking (Fig. 1 vs. Fig. 3), i.e., the ARWU ranking seems to be more stable. This is on one hand a good message: a smaller amount of fluctuations, but on the other hand, it has to be asked if a yearly publication of the ARWU makes sense, if no "real" changes can be expected. However, the same holds true for all rankings published on a yearly basis: no factual changes reflecting university strategies can be expected.

The astonishing low correlation between the ranks of the THE and the ARWU ranking, particularly for the universities ranked below 50 in both rankings, creates another serious doubt if rankings should be used for any management purposes at all. Maybe a "meta-analysis" of rankings could be reasonable to derivate consistent and reliable results from

rankings. If done, such and meta-analysis should include as many rankings as possible to reduce the amount of inconsistent fluctuations that do not reflect actual university performance.

## CONCLUSION

Both rankings show fluctuations in the rank and score particularly for universities with lower ranking position (below position 50) which lead to inconsistent "up and downs" in the total results, especially in the THE and to a lesser extend also in the ARWU. The observed fluctuations do most likely not correspond to academic performance, therefore neither the THES nor the ARWU offer great value as a tool for university management in their current forms for universities ranked below 50.

### Funding
The authors declare there was no funding for this work.

### Competing Interests
The authors declare there are no competing interests.

### Author Contributions
- Johannes Sorz and Martin Fieder conceived and designed the experiments, performed the experiments, analyzed the data, wrote the paper, prepared figures and/or tables, reviewed drafts of the paper.
- Bernard Wallner and Horst Seidler wrote the paper, reviewed drafts of the paper.

### Data Availability
The following information was supplied regarding the deposition of related data:

All raw data can be accessed publicly at the following web pages:.

THES— https://www.timeshighereducation.co.uk/world-university-rankings/2015/world-ranking/#/

ARWU—http://www.shanghairanking.com/de/ARWU2014.html.

### Supplemental Information
Supplemental information for this article can be found online at http://dx.doi.org/10.7717/peerj.1217#supplemental-information.

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
