# Peer review of "Inconsistent year-to-year fluctuations limit the conclusiveness of global higher education rankings for university management"

_PeerJ, doi:10.7717/peerj.1217_

## Round 0.1 · original submission · Major Revisions

Please, if possible, try to answer all the concerns raised by the reviewers. In addition, please modify the title to conform to the concern of reviewer 2 about the use of the statistical terminology (noise). Is it also possible to describe how the scores are calculated?

Reviewer 1 ·

Basic reporting

The manuscript is well written.

Experimental design

This paper describes regression analyses on THES and ARWU ranking from 2010 to 2014.

Validity of the findings

The authors conclude that the observed fluctuations in the THES do not correspond to actual university performance and ranking results are thus of limited conclusiveness for the university management of lower scoring universities. And they suggest that THE and ARWU alter their ranking procedure insofar as universities below position 50 should be ranked summarized only in groups of 25 or 50. I have some concerns on the methodology and analysis.

Additional comments

This paper describes regression analyses on THES and ARWU ranking from 2010 to 2014. The authors conclude that the observed fluctuations in the THES do not correspond to actual university performance and ranking results are thus of limited conclusiveness for the university management of lower scoring universities. And they suggest that THE and ARWU alter their ranking procedure insofar as universities below position 50 should be ranked summarized only in groups of 25 or 50. The manuscript is well written. I have some concerns on the methodology and analysis.

1. The current analysis was limited to the universities that ranked from 1 to 200. It would be helpful to perform additional analysis on the low ranked universes that below 200 and assess the rank and score variation.

2. Besides the graphic illustration, the authors need to report the summary statistics such as regression coefficients and corresponding 95% confidence intervals for any pair-wised comparisons.

3. It would be interesting to show some results on the rank and score fluctuation along time from 2010 to 2014.

4. To better understand the THES and ARWU rank system, it would be helpful to provide a summary table with the detailed comparison scores between THES and ARWU.

5. The legend of Figure 3 is confusing. Should exclude “a) 2010; b) 2011; c) 1012; d) 2013; e) 2014.”

Reviewer 2 ·

Basic reporting

The manuscript follows largely the journal format, and includes adequate background introduction and rationale for the objectives.

The figures presented are relevant to the content; however, they can be better labeled
(i) The captions appear to be rather non-specific and inconsistent. For example, line 293 “Figure …. from the taking 2010-11 on.” could have been given specific years, as in lines 310-311.
(ii) Also, for Figure 3, line 305, “(a-e) Ranks..” seems to be incorrect. The authors are advised to revise the captions, esp. for figures 1-4, for accuracy and consistency.
(iii) For Figure 5, the y-axis could be labeled as “ARWU raking 2014” corresponding to the x-axis “THES Ranking 2014”. It would be helpful to provide the number of universities used for the plots in the caption.

One minor comment: The abbreviation “THES” should be used throughout; “THE” should be changed in lines 76 and 86.

Experimental design

The data are graphically summarized in figures, which form the basis of the authors’ conclusions. It would have been helpful to provide quantitative evaluations of the observed differences.

Validity of the findings

The authors’ usage of the statistical terms, such as noise and random, differs from the conventional definitions. Specifically, (quoting from en.wikipedia.org) “statistical noise is the colloquialism for recognized amounts of unexplained variation in a sample”, and the observed fluctuations discussed in the manuscript could be due to the dynamic changes (explained) for the areas included (weighted) for deriving the overall scores. Fluctuations could be examined through regression analyses, while accounting for explained variations, e.g. publication increases/decreases related to faculty recruits/retirements, and increased teaching due to student enrollment expansion. The authors are advised to revise the terminology.

The authors describe the figures as regressing the x-axis (e.g. Figure 1, 2011-2012, year t-1) on the y-axis (e.g. 2011-2012, year t)—the common practice is to regress y on x (i.e. switch the two axes). It’s worth mentioning that the two regressions are not the same, though correlation between x and y remains the same.

Additional comments

The results presented in the manuscript are expected, given the inherent difference between the two indices, which aim to capture multiple dimensions of “performance” . The authors’ recommendation of using ordinal groups for ranks below 50 or 100 seems sensible.

---

## Round 0.2 · accepted · Accept

The comments were appropriately addressed.

Reviewer 1 ·

Basic reporting

The revised manuscript is much improved and address all my issues and suggestions for additional analyses.

Experimental design

Appropriate

Validity of the findings

Appropriate

Additional comments

The revised manuscript is much improved and address all my issues and suggestions for additional analyses.